# Foraging Patterns of Non-Territorial Eastern Imperial Eagle (*Aquila heliaca*): A Case of Successful Adaptation

**Dimitar Demerdzhiev** [1,2,*], **Ivaylo Angelov** [1,2,†] **and Dobromir Dobrev** [1]

1 Bulgarian Society for the Protection of Birds/BirdLife Bulgaria 5, Leonardo da Vinci Str., 4000 Plovdiv, Bulgaria
2 National Museum of Natural History, Bulgarian Academy of Sciences, 1000 Sofia, Bulgaria
* Correspondence: dimitar.demerdzhiev@bspb.org or dimitar.demerdzhiev@gmail.com
† Present address: Nature Conservation Centre Eastern Rhodopes, Madzharovo, 6300 Haskovo, Bulgaria.

**Abstract:** The Optimal Foraging Theory predicts that, to maximize fitness, animals adapt their foraging strategy that provides the most benefit for the lowest cost, maximizing the net energy gained. While the diet of many breeding raptor populations is well known, studies on the foraging patterns of non-territorial birds of prey (floaters) are scarce. In this study, we examined the foraging pattern of non-territorial Eastern Imperial Eagle, scrutinizing different aspects of its feeding ecology and behavior. We built a simple model of the optimal foraging strategy of floater eagles including the success of foraging as a currency as well as environmental factors such as seasons, type of prey, habitat, foraging techniques, and eagle age as a limitation affecting the foraging efficiency of birds. We found that floaters focused their diet exclusively on European Souslik, accounting for almost half (44.2%) of the eagle's prey. Diet differences between floaters and breeders were due to higher Souslik and carrion consumption and lower Hedgehog predation by floater eagles. The diet diversity of breeding eagles (H = 3.297) was much higher than that of floaters (H = 1.748). Our model suggested that the foraging mode, habitat type, and season best explained the feeding success of non-territorial eagles ($_\Delta$AIC = 0.00, $w$ = 0.42). Of all explanatory factors, "Kleptoparasitism" ($\beta^2$ = −4.35), "Rodents" ($\beta^2$ = −4.52), "Pasture" ($\beta^2$ = 2.96), "Wheat" ($\beta^2$ = 4.41), "In the air" ($\beta^2$ = 4.16), and "Other habitats" ($\beta^2$ = 4.17) had a pronounced effect. The factors "Spring–summer season" ($\beta^2$ = −0.67) and "European Souslik" ($\beta^2$ = −2.76) had a marginal effect in our models. Generally, the mean success rate of attack modes used by non-territorial eagles was 0.54 ± 0.50. Floaters successfully obtained food through: kleptoparasitism (43.10%), carrion feeding (24.14%), and high soar with vertical stoop (14.66%). Several important issues for the conservation of non-territorial Eastern Imperial Eagles arose from our research. The strong relation of floaters with the European Souslik calls for specific conservation measures aimed at the conservation of this type of prey and the restoration and appropriate management of its grassland habitats. The importance of the scavenging behavior of juvenile birds requires increased control of the use of poison baits and subsequent prosecution by state institutions. Protecting the most important temporary areas, improving institutional control against the use of poison baits, and intensifying awareness-raising campaigns among pigeon-fanciers and hunters are also of crucial importance for effective species conservation.

**Keywords:** top predator; floaters; diet; raptors; feeding ecology; behavior; conservation; temporary areas

## 1. Introduction

The classic Optimal Foraging Theory (OFT) predicts that, to maximize fitness, animals adapt their foraging strategy that provides the most benefit (energy) for the lowest cost (time/effort), maximizing the net energy gained [1,2]. However, new concepts, such as balancing between foraging and safety, the assumption that tactical foraging decisions depend on state variables, such as fat reserves (state dependence theory), foraging games

(game theory), and the consequences of foraging in a group, are incorporated in the decisions the animal must make [3]. The new aspect of the feeding behavior concerns the physiological, biochemical, and anatomical mechanisms that can constrain an animal and thus influence its foraging actions. Foraging behavior is crucial with regard to evolutionary biology not only because it is a major factor in the survival, growth, and reproduction of animals but also because of the resulting adaptations that persist in the course of evolution. This complex process is influenced by numerous factors from cognitive and physiological limitations to predation and social interactions [3,4].

While modeling foraging behavior, it should be kept in mind that organisms maximize a variable known as the currency, a unit including costs and benefits that are imposed on the animal. The constraints of the environment are key factors that can limit the forager's ability to maximize the currency. Then, the organism's best foraging strategy is defined as the decision that maximizes the currency under the constraints of the environment [5]. In most species, the availability and accessibility of food resources have been identified as the key factors that shape foraging behavior and dietary decisions [2,6]. Raptors can adapt their foraging as a response to main prey depletion [7] or to avoid competition [8].

The food spectrum and dietary relationship with breeding are well known for different diurnal and nocturnal birds of prey [9–14]. However, studies on the foraging patterns of non-territorial birds (floaters) of raptor species are scarce [15–19]. For many raptors, survival during the dispersal period has important consequences for the population trajectories in the future ( [20,21]. During this period, the floaters of some large eagles tend to restrict their movements to a few favorable places, known as "temporary settlement areas" [20,22,23]. These sites are normally outside of the breeding territories and are characterized by the abundance of prey, where floaters spend periods of varying duration before joining breeding populations [15,22,24]. However, this differs from the juvenile high-mobility strategy demonstrated by tropical raptors [25].

The Eastern Imperial Eagle (*Aquila helica*) (hereafter EIE) is a large raptor species distributed from Eastern Europe to Central Asia [26,27]. This open-ground eagle forages in different habitats, where it exploits diverse prey such as Sousliks (*Spermophilus* sp.), Leporids (*Leporidae*), Hedgehogs (*Erinaceus* sp.), Corvids (*Corvidae*), Gulls (*Larus* sp.), White Stork (*Ciconia ciconia*), and various Reptiles (*Squamata, Testudinidae*) [10,11,28–34]. While the study of the diet of breeding EIEs is well documented throughout the distribution range of the species [10,30–32,34], the foraging of floaters is only known from sporadic observations [19], and there is no systematic survey presenting the various aspects of the foraging pattern of non-territorial birds. Although the populations of EIE in Europe are stable or even increasing in some parts of the distribution range [10,26], the species' global population is considered to be declining [35]. The species suffered from habitat loss and alteration, electrocution by power lines, poison baits, direct persecution, and prey decrease affecting both breeding territories and dispersal areas [31,36–38]. Therefore, preserving and securing appropriate settlement areas [21], along with the conservation of breeding grounds [36], is crucial for EIE population viability.

In this study, we examined the foraging patterns of non-territorial EIEs, exploring different aspects of their feeding ecology and behavior. By searching relationships between the success of feeding and the factors limiting/supporting it, we built a simple model of floater EIEs foraging including the success of foraging as a currency and environmental factors such as seasons, type of prey, habitat, foraging techniques, and eagle age as a constraint limiting the foraging efficiency of eagles. We predicted that: (1) eagles used different techniques to obtain food at different ages, and adults were more experienced and more successful in hunting, while less experienced juveniles and immature individuals could have a lower hunting success and, as a result, used alternative techniques (kleptoparasitism, scavenging) to optimize foraging; (2) habitat type, vegetation height, seasons, and type of prey influenced foraging strategy and success. We compared our data on floaters' diet with the information available for the breeding population and hypothesized that the diet of non-territorial eagles differed from the food of territorial birds and that this dietary

diversification could be considered an example of successful adaption driven by evolution. Based on our findings, we discussed and recommended conservation efforts that should be taken to preserve and secure important settlement areas. Our results can be applied for other threatened large raptor species using the same foraging strategy and behavior during the dispersal period.

## 2. Materials and Methods

### 2.1. Study Design

We collected data from dispersal areas (Figure 1) of EIE in Bulgaria over a period of 25 years (1998–2022). A total of 186 cases of the foraging of non-territorial EIEs were documented, of which only 7 (3.76%) of the attempts were of unknown capture success due to the distance from the observer and/or local topography. Most of the observations (n = 147; 79%) were made in two important temporary settlement areas, Besaparski hills and Sliven field, which harbored dozens of floaters yearly [24,39]. These two areas were very similar and consisted of karst hills with almost no vegetation, as well as thermophilic grass communities dominated by Bluestem (*Dichanthium ischaemum*), Scented grass (*Chrysopogon gryllus*), and Needle grass (*Stipa capilata*), imparting the steppe character of the habitats [40]. In addition, we used the data (food remains; pellets, n = 22) of 20 identified preys of three EIEs equipped with satellite transmitters (PPT). Having located the roost sites of the birds, we visited the area and collected food remains and pellets.

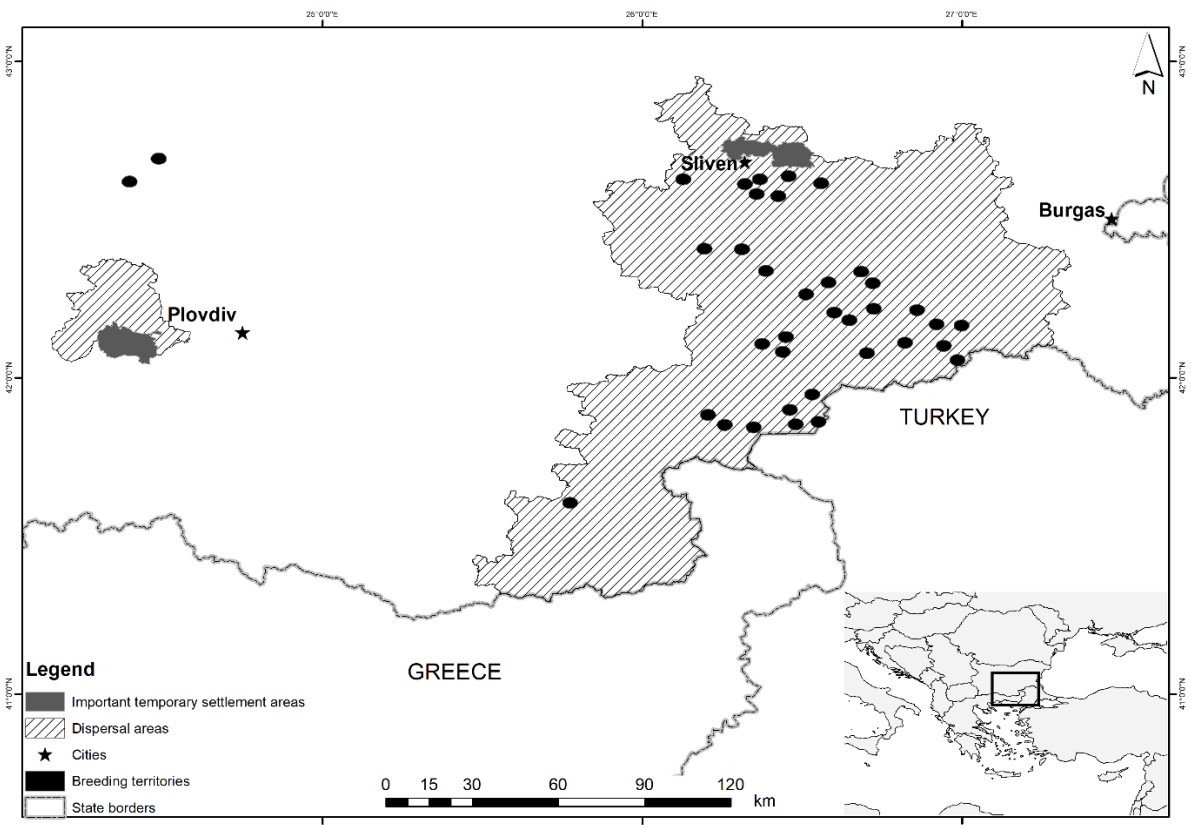

**Figure 1.** Dispersal areas (shaded) and breeding territories (black dots) used by the Eastern Imperial Eagle (*Aquila heliaca*). Two important temporary settlement areas (Besaparski hills and Sliven field) are given in grey.

Observations were made during the whole study period, which were particularly intense in both important settlement areas (Besaparski hills and Sliven field) during the period 2002–2011. During these years, the two sites were visited either two or three times per month, with an observation duration of two to eight hours. In the rest of

the studied period, the two places were visited once every two or three months. The remaining observations were made in dispersal areas of secondary importance throughout the entire years. All observations in the study accounted for >1700 h of field work. For each observation, we noted the following information: day, hour, individual, eagle age, habitat characteristics, hunting or feeding technique used, success of the feeding/hunting technique, and prey [16]. Eagle plumage was identified according to [41]. For the purpose of the analyses, we grouped the different ages of eagles into three classes: juvenile (first calendar autumn—second calendar summer); immature (between second calendar autumn and sixth plumage); and adult (more than seven calendar autumn). An attack was defined as a direct attempt to capture/steal clearly identifiable prey [42], and a capture was an attack that resulted in the acquisition of prey by an eagle [43]. Thus, each attack was classified either as a success or a failure. Observations were made from points offering good visibility and consisted in active scans to detect the predator and the prey within a radius of 400–500 m around the observer [17]. All capture attempts with undetermined outcomes were excluded from the analysis [44]. However, all the observations of carcass feeding (n = 28) were considered successful because, in all cases, the individual accessed the carrion source [16].

The type of attack was defined by the position of the eagle at the beginning of the attack. Hunting behavior was determined according to [45,46] with some additions. We categorized five different hunting techniques to obtain food (Table 1). The other foraging techniques were separated in single categories. Six variables were evaluated for each attack (Table 1).

**Table 1.** Definition of the variables used in models to analyze the attack success and feeding behavior of Eastern Imperial Eagles.

| Variable | Variable Type | Description |
| --- | --- | --- |
| Age | Categorical | Three age-classes: adults, immature, and juveniles. |
| Attack/Foraging Mode | Categorical | Ten classes: (1) Powered contour flight; (2) High soar with vertical stoop and descent attack; (3) Glide attack with tail-chase; (4) High-perch; (5) Walk-grab; (6) Collect a crashed animal; (7) Kleptoparasitism; (8) Carrion feeding; (9) Cooperative feeding; (10) Unspecified |
| Prey type | Categorical | Prey species. The prey's single specimens are grouped into a common category. Nonidentified prey was given in a separate category. |
| Habitat type | Categorical | Habitat characteristics according to land use pattern: (1) Pasture; (2) Stubble; (3) Wheat; (4) Fallow; (5) Other, including single casses such as asfalt road, ekoton, quarry, shrubs, and fishpond;.(6) The airstrikes were divided into a separate category: "in the air" |
| Vegetation height | Continuous | The height of the vegetation in cm. |
| Seasons | Categorical | Two categories with an equal duration: (1) Spring–summer (from March to August) and (2) autum–winter (from September to February) |

### 2.2. Statistical Analyses

We carried out a non-parametric Chi-square test with Monte Carlo randomization (9999 permutations) [47] to compare the diet of floaters in dispersal areas and breeding populations [15]. The data about the breeding EIE's diet were taken from the available literature [10]. However, the breeding population and the studied dispersal territory were distributed in the same area, with a maximum distance of 30 km between the dispersal places and breeding pairs (Figure 1). Therefore, we assume that our dietary data were geographically independent. Although different methods were used to collect the information (analysis of food remains and pellets for the breeding population vs. mostly visual observations of diet patterns of floaters), the two surveys were conducted simultaneously and covered the same annual/monthly periods. Therefore, the restrictions arising from the

different methods are not significant and describe the general dietary patterns of the two sections of the population (breeders and floaters).

Food diversity calculated with the Shannon–Weaver index (H) [48] was also used in the diet analysis of the breeding population [10]. We applied Abundance-based Coverage Estimator (ACE) [49] for the bias correction (bootstrap 9999 simulations) of H. This corresponded to the Bias-corrected Maximum Likelihood Estimator (MLE) for Shannon's index given by [50].

To estimate the success rate of different foraging strategies used by eagles, we built a simple model using Generalized Linear Mixed Models (GLMM) with a binomial error distribution and logit link function. Foraging success was modeled as a binary variable (1 = success, 0 = failure). The constraint factors of foraging success included in the model as response variables were: season, age class of eagles, foraging mode, prey type, habitat type, and vegetation height (Table 1). Due to the impossibility of separate birds being individually identified and tracked over time, we did not include the "individual eagle ID" as a random factor in our model. Thus, the effect of the "individual" was not evaluated. We used the Akaike Information Criterion corrected for small sample sizes (AICc) for model selection and chose the models with the lowest $AIC_c$ value from the set of our candidate models. All models with an $AIC_c$ value < 2 from the model with the lowest $AIC_c$ ($AIC_{cmin}$) were considered the best models ($_\Delta AIC_c = AIC_i - AIC_{cmin}$) [51]. The relative importance of each model was estimated through the weight of AICc ($w$), so all of the weights for all models added up to 1. Explanatory parameter estimates ($\beta^2$) with Lower (95%) and Upper CL (95%) and a probability value ($p$) of the explanatory factors were also evaluated.

All data were analyzed using Statistica for Windows, Release 12 [52], R v.2.15.2 [53], and Past Version 4.08 [54]). Results with $p \leq 0.05$ were considered significant. Values were provided as the means $\pm$ SE.

## 3. Results

### 3.1. Diet Diversity and Comparison of Floaters vs. Breeders

We found that floaters based their diet exclusively on European Souslik (*Spermophilus citellus*), accounting for almost half (44.2%) of the eagle's prey (Table 2). The other important food sources for non-territorial eagles were carrion (21.1%), followed by Small Rodents (Rodentia, excl. Souslik) (7.25%), Feral Pigeon (*Columba livia f. domestica*) (6.52%), and Brown Hare (*Lepus europaeus*) (3.62%). Other prey such as Rook (*Corvus frugilegus*), Golden Jackal (*Canis aureus*), Snakes (Serpentes), Common Kestrel (*Falco tinnunculus*), and Northern White-breasted Hedgehog (*Erinaceus roumanicus*) appeared occasionally (Table 2). Non-detected prey accounted for 10.14% of the consumed animals.

**Table 2.** Diet of non-territorial Eastern Imperial Eagle in dispersal areas.

| Prey | Observation (ind.) | Food Remains (ind.) | Total (ind.) | Total (%) |
|---|---|---|---|---|
| *Spermophilus citellus* | 50 | 11 | 61 | 44.20 |
| Rodentia (excl. Souslik) | 10 | | 10 | 7.25 |
| *Columba livia f. domestica* | 1 | 8 | 9 | 6.52 |
| *Lepus europaeus* | 5 | | 5 | 3.62 |
| *Serpentes* | 1 | 1 | 2 | 1.45 |
| *Falco tinnunculus* | 1 | | 1 | 0.72 |
| Carrion | 29 | | 29 | 21.01 |
| Unidentified | 14 | | 14 | 10.14 |
| *Corvus frugilegus* | 4 | | 4 | 2.90 |
| *Erinaceus roumanicus* | 1 | | 1 | 0.72 |
| *Canis aureus* | 2 | | 2 | 1.45 |
| **Total** | **118** | **20** | **138** | **100** |

Prey frequency in the diet significantly differed between floaters (in dispersal areas) and breeders (territorial pairs) due to higher Souslik ($\chi^2$ = 9.94, df = 1, Monte Carlo: $p$ = 0.002) and carrion consumption ($\chi^2$ = 16.22, df = 1, Monte Carlo: $p$ < 0.001) and lower Hedgehog ($\chi^2$ = 23.62, df = 1, Monte Carlo: $p$ < 0.001) predation by floater eagles (Figure 2). The diet diversity of breeders (H = 3.297) was much higher than that of floaters (H = 1.748).

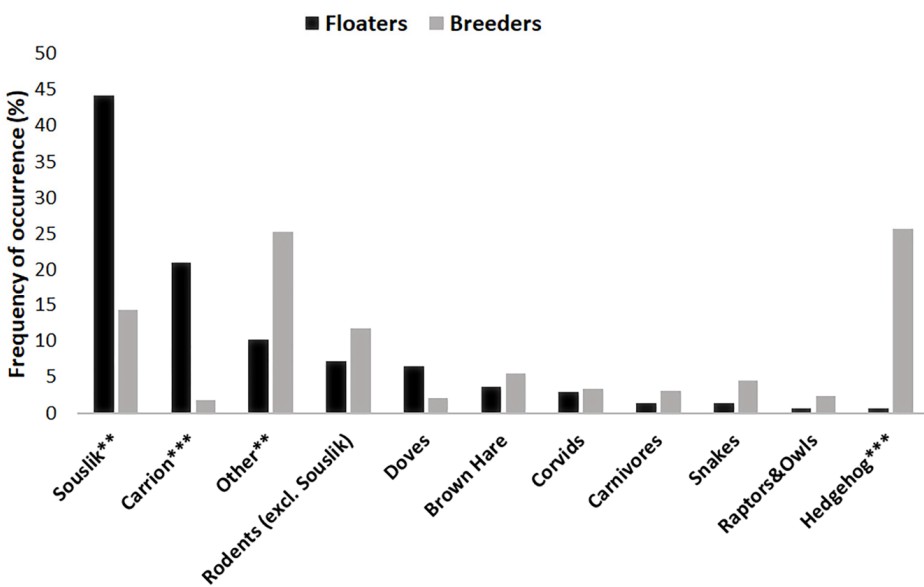

**Figure 2.** Comparison of prey occurrence in the diet of floaters and breeders of Eastern Imperial Eagle in Bulgaria. Data for breeders (territorial pairs) were taken from [10]. Significant values are given in *.

*3.2. Foraging Pattern of Non-Territorial EIEs*

By modeling the foraging pattern of non-territorial eagles, we found that "Seasons" + "Foraging Mode" + "Habitat type" primarily explained successful feeding ($_\Delta$AIC = 0.00, $w$ = 0.42). The second-ranked model included only "Foraging Mode" and "Habitat type" ($_\Delta$AIC = 1.39, $w$ = 0.21), while the third one included "Prey type" + "Habitat type" ($_\Delta$AIC = 1.50, $w$ = 0.20) (Table 3). Of the explanatory factors, "Kleptoparasitism" ($\beta^2$ = −4.35, Wald. Stat. = 6.03, $p$ = 0.01) and habitats had a pronounced effect (Table 3). Factors such as "Spring–summer season" ($\beta^2$ = −0.67, Wald. Stat. = 3.82, $p$ = 0.051) and "European Souslik" ($\beta^2$ = −2.76, Wald. Stat. = 3.87, $p$ = 0.049) had only a marginal impact in our models.

**Table 3.** List of GLMMs used for the analysis of the foraging pattern of non-territorial Eastern Imperial Eagle. All models with $_\Delta$AIC < 2 were considered the best models. The degree of freedom (df), model weight value ($w$) and probability value of each model ($p$) were also given. Parameter estimates ($\beta^2$) ± SE, Lower (95%) and Upper CL (95%) of significant explanatory factors, their importance value (Wald Stat.), and a probability value ($p$) were taken from the average model.

| N | Model Structure | AIC | $_\Delta$AIC | df | $w$ | $p$ |
|---|---|---|---|---|---|---|
| 1 | Seasons + Foraging Mode + Habitat type | 175.77 | 0.00 | 13 | 0.42 | <0.001 |
| 2 | Foraging Mode + Habitat type | 177.16 | 1.39 | 12 | 0.21 | <0.001 |
| 3 | Prey type + Habitat type | 177.26 | 1.50 | 20 | 0.20 | <0.001 |
| 4 | Seasons + Foraging Mode + Habitat type + Vegetation (cm) | 177.43 | 1.66 | 14 | 0.17 | <0.001 |

| N | Explanatory factors | $\beta^2$ | St.err. | Lower CL/Upper CL | Wald Stat. | $p$ |
|---|---|---|---|---|---|---|
| 1 | Spring–summer season | −0.67 | 0.34 | −1.34/0.00 | 3.82 | 0.051 |
| 2 | Kleptoparasitism | −4.35 | 1.77 | −7.81/−0.88 | 6.03 | 0.01 |
| 3 | European Souslik | −2.76 | 1.40 | −5.51/−0.01 | 3.87 | 0.049 |
| 4 | Rodents (excl. Souslik) | −4.52 | 1.78 | −8.02/−1.03 | 6.43 | 0.01 |
| 5 | Pasture | 2.96 | 0.54 | 1.90/4.02 | 30.07 | <0.001 |
| 6 | In the air | 4.16 | 0.80 | 2.60/5.72 | 27.29 | <0.001 |
| 7 | Other habitats | 4.17 | 1.05 | 2.12/6.22 | 15.87 | <0.001 |
| 8 | Wheat | 4.31 | 0.67 | 3.00/5.62 | 41.46 | <0.001 |

### 3.3. Foraging Mode and Success

Non-territorial EIEs successfully obtained food through: kleptoparasitism in 50 (43.10%) of the cases, high soar with vertical stoop in 17 (14.66%) of the successful attacks, walk-grab in 5 (4.31%) of the cases, cooperative feeding and high-perch in 2 cases each (1.72%), crashed animal pickup in 1 case (0.86%), and carrion feeding in 28 (24.14%) of the cases. In 11 (9.48%) of the cases, the foraging mode was unspecified.

The most successful attack techniques were walk-grab (n = 8 cases; 62.5% success) and kleptoparasitism (n = 81 cases; 61.73%), followed by high-perch (n = 4 cases, 50% success) and high soar with vertical stoop (n = 36 cases; 47.22% success). However, the hunting techniques of a proven unsuccessful rate were: glide attack with tail-chase (n = 7 cases) and powered contour flight in one case.

### 3.4. Effect of Habitat Type, Prey Type, and Season

As our global model demonstrated, pasture (n = 80), air habitat (n = 51), and wheat (n = 23) were the most common habitat types used by eagles. Fallow (90%), pasture (78.75%), and stubble (75%) were the habitats most successfully used for foraging by birds, while airily environments were mostly unsuccessful (60.78%). The attack success in wheat and other habitats was almost equal to unsuccessful attempts (Figure 3).

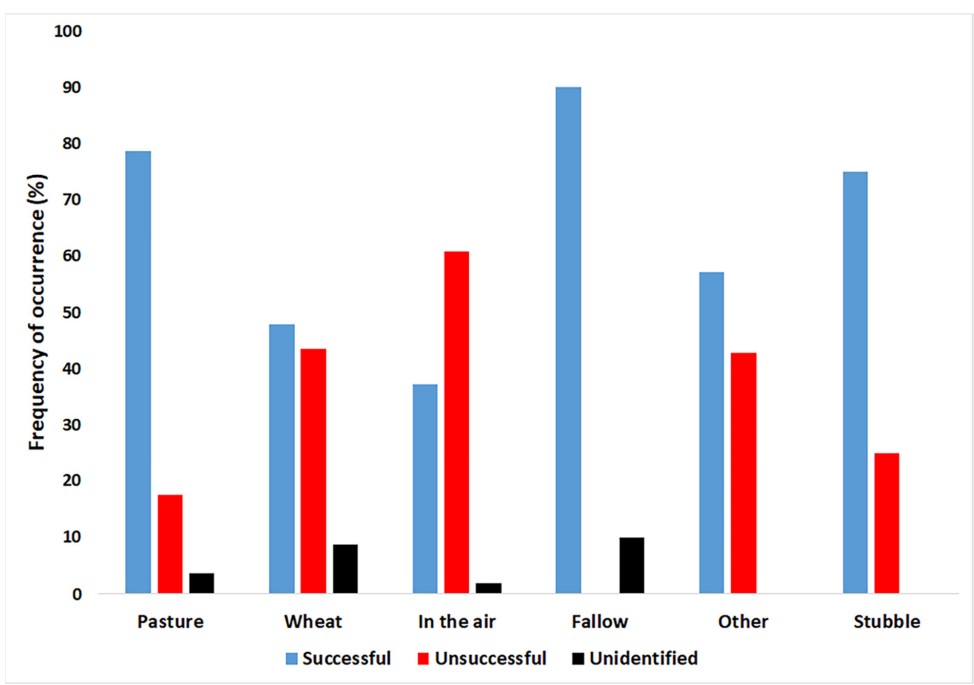

**Figure 3.** Results of the foraging of non-territorial Eastern Imperial Eagles in different habitat types.

In the spring–summer season, non-territorial EIEs exclusively exploited European Souslik (73.61%), while in the autumn–winter period, they consumed primarily carrion (42.31%) (Figure 4). However, in winter, Feral Pigeon ($\chi^2$ = 11.66, df = 1, Monte Carlo: $p < 0.001$), Brown Hare ($\chi^2$ = 4.37, df = 1, Monte Carlo: $p = 0.008$), and Rodents ($\chi^2$ = 4.77, df = 1, Monte Carlo: $p = 0.01$) significantly increased their frequency in the eagle's diet (Figure 4). Regarding the active hunting techniques, the eagles used glide attack with tail-chase more frequently in winter ($\chi^2$ = 5.54, df = 1, Monte Carlo: $p = 0.004$), while high soar with vertical stoop was used mostly in summer, although this difference was marginal ($\chi^2$ = 3.05, df = 1, Monte Carlo: $p = 0.06$). However, 59.30% of the active hunting modes were successful in summer, while only 45.10% of the attacks benefited in the winter season.

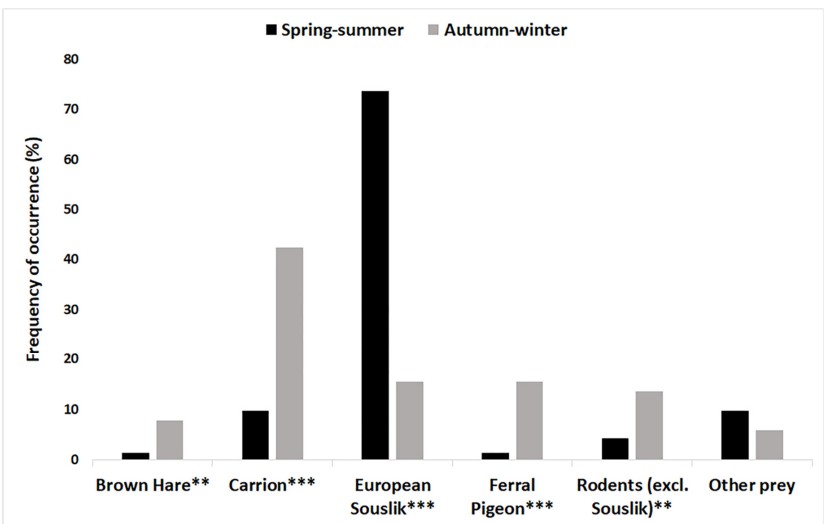

**Figure 4.** Frequency of occurrence (%) of different prey types used by non-territorial Eastern Imperial Eagles in different seasons. Significant values are given in *.

*3.5. Effect of Eagle Age*

As a whole, the mean success rate of attack modes used by non-territorial eagles was $0.54 \pm 0.50$ (n = 138). Corresponding with our "Global model", eagle age did not influence the success of attacking modes (Kruskal–Wallis test: $H_2 = 1.54$, $p = 0.46$). The mean success rate of adults was $0.48 \pm 0.11$ (n = 23). Immature eagles successfully attacked/stole prey with mean rate of $0.57 \pm 0.05$ (n = 105), while for juveniles, this rate was $0.40 \pm 0.16$ (n = 10). Surprisingly, juvenile eagles used kleptoparasitism in only 25% of the documented foraging (n = 20), and this significantly differed ($\chi^2 = 8.33$, df = 1, Monte Carlo: $p = 0.004$) from the registered cases for immature eagles (50%, n = 130) and adults (48%, n = 25) (Figure 5). Expectedly, juveniles relied mainly on carrion feeding (50% of foraging behavior), and this differed significantly ($\chi^2 = 21.61$, df = 1, Monte Carlo: $p < 0.0001$) from the scavenging practices of immature eagles (13.08%) and adults (4%) ($\chi^2 = 39.19$, df = 1, Monte Carlo: $p < 0.0001$). High soar with vertical stoop was the most often used active hunting technique for all age classes (Figure 5).

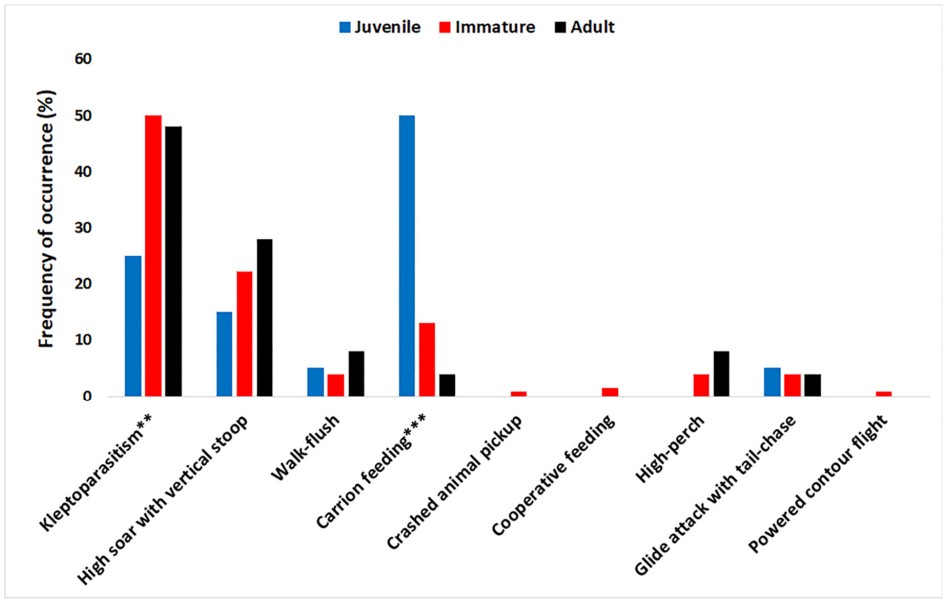

**Figure 5.** Frequency of occurrence (%) of different foraging modes used by different age classes of non-territorial Eastern Imperial Eagles. Significant values are given in *.

## 4. Discussion

### 4.1. Dietary Diversification as a Case of Successful Adaptation

Little is known about the foraging strategy and diet differences between floaters and territorial birds of large raptor species and how they choose prey and perform feeding behavior (however, see [15]). It is well known that the availability and accessibility of food resources are key factors that shape the foraging behavior and dietary choice [2,6]. Top predators can adapt their diet mainly in response to habitat alteration and the depletion of main food resources [7,55,56], but also to avoid competition [8]. Our hypothesis that the diet of non-territorial EIE differed significantly from the food choice of breeding birds was fully confirmed. Floater birds displayed dietary preferences for European Souslik, which largely determined their dispersal pattern and foraging strategy. In their dispersal grounds, they found temporary areas, where Souslik abundance existed, and where they formed concentrations by dozens. Searching for food, non-territorial eagles moved from site to site, sometimes covering great distances, where they opportunistically used any readily available food source such as carrion. Specialization in Souslik as a keystone prey led to low trophic diversity in non-territorial eagles. Sousliks had become ideal prey for inexperienced floater eagles, since they can be locally abundant, offering high energy value and low hunting costs. Consequently, floater EIEs depended heavily on Sousliks, and this would explain their ecology and behavior during the dispersal phase. In contrast, breeding EIEs depended on the type of prey in their territories, and when there were not enough Sousliks, they had to make a trade-off decision about whether to occupy that territory and adapt to other less profitable prey species, such as hedgehogs, or harder to catch prey, such as storks, or seek another territory, behaving like floaters, but this could hardly be proven. With an insufficient abundance of the main prey in the breeding territory, eagles exploited various type of caught prey and thus had a more diverse diet [10]. Similar diet diversification between floaters and breeders was found for another top predator: the Bonelli's Eagle (*Aquila fasciata*) [15].

We speculate that this diet difference between floaters and breeders represents an adaptive mechanism to avoid intraspecific competition in populations of large raptor species. Floaters, which are mostly younger and less experienced individuals, are attracted to places that have an abundance of easier-to-obtain and more profitable prey, such as Sousliks, usually away from territory defenders and non-tolerant breeders. Here, floaters more easily survive, increasing their skills, which would help them capture more difficult and diverse prey when they occupy a territory in the future. This phenomenon in the life history of eagles is a process probably driven by evolution, but deeper insight is needed.

### 4.2. Factors Influencing Foraging Behavior and Success

Our model suggests that foraging mode, habitat type, and season best explain feeding success, followed by prey type. Kleptoparasitism was the most effective hunting technique used by non-territorial EIEs, as almost half of the successful hunts consisted of stealing the prey from another predator. Contrary to our expectations, juvenile eagles quite rarely used kleptoparasitism as a foraging mode, unlike immature and adult birds, and this differed from the findings of another study on the Spanish Imperial Eagle (*Aquila adalberti*) [16]. Apparently, in a hierarchical relationship, juveniles are less experienced than older and more powerful eagles; hence, they try to steal prey from them less often, resorting exclusively to feeding on carrion. Conversely, more experienced and more suspicious immature and adult birds resort to the riskier and more atypical scavenging practices less often and use kleptoparasitism or active hunting. We consider that the differences in the findings of the Spanish Imperial Eagle study are due to the different methodological approach. Margalida et al. [16] compared the foraging mode of eagles of different age classes, including breeders and floaters, while our study was focused only on non-territorial birds. However, kleptoparasitic behavior is widespread among birds [57,58] and common among raptors [59]. The use of kleptoparasitism by floaters as a main foraging technique to obtain food probably differs from the foraging behavior of territorial eagles. Breeders must defend a territory,

engage in mating behavior, and raise offspring, and these factors are more important in terms of maximizing fitness [2,60]. They have to minimize the time spent on foraging and probably do not have enough time, unlike floaters, which can sit and watch or soar for hours and wait for a convenient opportunity to steal prey from another predator. Then, kleptoparasitism seems to be a low-cost, highly profitable foraging mode in floater eagles. Anyway, this issue needs further clarification.

High soar with vertical stoop was the active hunting technique most frequently used by the eagles, yielding success in almost half of the cases. Birds used high soar primarily during the summer period, when ascending thermals particularly favored this type of prey searching. However, this hunting technique is widespread among many diurnal birds of prey [27]. Through walk and grab, eagles successfully obtained medium-sized or small prey such as Sousliks or voles. This hunting technique was frequently used by another eagle such as the Lesser Spotted Eagle (*Clanga pomarina*) in areas with a high prey density [61]. Glide attack with tail-chase was used by eagles exclusively during the winter period, when they tried to catch agile and maneuverable prey, such as pigeons, in the air. The predator's poor position when starting the attack, as well as the lack of surprise, were probably the reasons why all these attempts ended in failure. We found that the other hunting strike techniques, such as high-perch and powered contour flight, were rarely used by non-territorial EIEs. High-perch was a common hunting mode used by different large raptors such as the White-tailed Eagle (*Haliaeetus albicilla*), Spanish Imperial Eagle, Lesser Spotted Eagle, and Bonelli's Eagle [16,17,62,63]. However, powered contour flight with short glide attack and walk-grab were successfully used by another species closely related to EIE: the Steppe Eagle (*Aquila nipalensis*) [27].

In line with our expectations, the habitat type strongly affected the foraging success of eagles. This was in agreement with several studies that had shown that the landscape characteristics and prey types were also factors influencing hunting success [16,17,63]. While pasture, fallow, and stubble were the most successfully used habitats, the air environment was mostly unsuccessful. Being larger and less mobile, the EIE hunts mostly terrestrial prey dominated by various medium-sized and small mammals and reptiles. Birds on the ground are also easier to catch, especially when taken by surprise. Back in the air, eagles are not as agile and fast as hawks or falcons, and hunting there more often ends in failure. An important fact to consider here is that non-territorial eagles are mostly juvenile and immature birds, and hunting airborne prey requires more skills and experience. The small differences in the shape and proportions of wings and tails between immature and adult eagles also affect the hunting ability [27]. Feeding on Storks or other more difficult aerial prey, such as gulls and pigeons, has been well documented for territorial EIEs [10,30]. Apparently, after gaining experience, eagles adapt to hunting more difficult-to-capture aerial prey. The individual abilities are not unimportant either. However, whether territorial eagles specialize in hunting birds by capturing them in the air or primarily on the ground is a question that needs further investigation.

Our expectation that vegetation height influenced the strike success of eagles did not find support in this study. We assume that this was due to the type of habitats used. The pastures, where the birds were primarily observed hunting, were well managed with a low grass height. The other used habitats, such as stubble or fallow, were also characterized by low vegetation or by a total absence of greenery. Interestingly, in tall and dense wheat, where it was more difficult to hunt, we saw almost equal success and failure in hunting. The lack of sufficient instances of hunting in taller vegetation was probably due to the avoidance of this type of landscape by the eagles. More light is needed, however, to clarify this issue.

Corresponding to other studies [10,63], our research demonstrated a clear relationship between seasons and the type of taken prey. In the spring–summer season, European Souslik predominated in the diet, while in the autumn–winter period, eagles tried to compensate for the lack of Sousliks by feeding on carrion, Ferral Pegion, Brown Hare, and Small Rodents. However, in winter, due to deteriorating weather conditions, the hunting

success was also lower. Similar results were found for another large top-predators such as the White-tailed Eagle [63].

The mean success rate of the attack modes found in our studies was similar to that recorded for White-tailed Eagle (50.5%) and greater than the one observed for Bonelli's Eagle (28.2%), Lesser Spotted Eagle (24%), and Golden Eagle (*Aquila chrysaetos*) (20%) [44,62,63]. Surprisingly, eagle age did not influence the success of attacking modes, and, at first sight, immature eagles had an even higher mean success rate than adults. This contradiction was due to the smaller sample size of adults foraging and the fact that immature eagles used mostly kleptoparasitism to obtain food. If we ignore kleptoparasitism as a phenomenon and consider only cases of active hunting, then the success of adult eagles would be 54.55%, and that of immature birds and juveniles would be 40%. This better corresponds to the other findings for different birds of prey [16,63–65]. Age-related improvements in foraging skills and experience benefit adults in using various hunting techniques, such as high soar with vertical stoop, walk-grab, and high-perch, more often than for the other age groups. An important fact to consider here is also the extent to which juvenile plumage morphology limits their ability to hunt the way that adults do [66]. The small differences in the length and stiffness of the remiges and rectrices between juveniles/immatures and adults put young birds at a disadvantage in terms of flying expertise [27]. Bulges of secondary feathers, shorter tails, and possibly softer remiges could hinder the ability of the juveniles to use high soar with vertical stoop and powered contour flight for hunting sousliks the way that adults do. However, neurological maturation and less developed pectoral muscles could also affect age differences in foraging behavior. Thus, further research is needed to better understand the age-related differences in the foraging behavior of large eagles.

### 4.3. Conservation Suggestion and Perspectives

Although some studies have investigated the diet of the breeding populations of EIEs in the entire distribution area [10,11,29–34], detailed reports during the dispersal phase are scarce, and this ecological trait during that particular life stage has not been considered in the management strategies for this threatened raptor.

Several important implications for EIE conservation emerge from our study. First of all, the strong dependence of non-territorial eagles on the European Souslik calls for specific conservation measures for this increasingly declining species [67]. There has been a trend for plowing and converting vast areas of grassland and semi-natural grass habitats into cropland over the last decade [68]. This process severely affects some of the most important places, both for the floater section of the EIE population and the breeding grounds. The floaters' tendency to form large concentrations in certain temporary settlement areas, where they spend long periods, requires the special protection of these places. Appropriate settlement areas may increase floaters' survival and guarantee population viability [21,69]. The recovery of Souslik populations through restocking [70], as well as the proper management of grassland habitats (through grazing by small animals such as sheep and goats) in these areas that are so important to non-territorial eagles, are of paramount importance for the conservation of the species. The significance of floaters to raptor population trajectories is well documented [21]. The most important places should be put under protection. The restoration of already damaged and plowed grassland habitats is also recommended.

Secondly, the importance of carrion for the survival of juvenile birds poses a severe risk of poisoning, due to illegal baits occasionally being used to control predators. Poisoning has been identified as the most important mortality factor affecting the breeding population of EIE in Bulgaria [37].

Thirdly, eagles feeding on pigeons can also raise conflicts with pigeon fanciers, which, in turn, could result in persecution incidents. Unfortunately, there have been such examples concerning non-territorial EIE [71].

Therefore, protecting the most important temporary settlement areas for non-territorial EIEs, restoring and subsequently properly managing damaged grassland habitats, strengthening the European Souslik through restocking programs, improving the monitoring

of important large and/or threatened souslik colonies, preventing poisoning incidents through increased control of the use of poison baits, as well as intensifying awareness campaigning among key stakeholders, such as pigeon-fanciers and hunters, are paramount to the conservation of non-territorial EIEs.

### 5. Conclusions

Our study demonstrates that floaters of a large top predator, such as the EIE, generally adapt their diet via Souslik dominance, thus avoiding intraspecies competition by breeders, and this prey dependence influences their dispersal pattern. Eagles can modify their foraging strategy to cope with variations in weather conditions and food availability. Our predictions that the different eagle ages involved different techniques to obtain food and that the habitat type and prey type influenced the foraging success found clear support. Kleptoparasitism was the most successful mode for non-territorials to obtain food, while glide attack with tail-chase was fully unsuccessful.

Several important issues for the conservation of non-territorial EIEs arose from our research. The strong relation of floater eagles to European Souslik calls for specific conservation measures aimed at the conservation of this prey and the restoration and appropriate management of its grassland habitats. The importance of the scavenging behavior of juvenile birds requires increased control over the use of poison baits and subsequent prosecution by state institutions. Increased awareness campaigning among pigeon-fanciers and hunters is also of crucial importance for effective EIEs conservation.

**Author Contributions:** Conceptualization, D.D. (Dimitar Demerdzhiev) and I.A.; methodology, D.D. (Dimitar Demerdzhiev) and I.A.; software, D.D. (Dimitar Demerdzhiev) and D.D. (Dobromir Dobrev); validation, D.D. (Dimitar Demerdzhiev), I.A. and D.D. (Dobromir Dobrev); formal analysis, D.D. (Dimitar Demerdzhiev); investigation, D.D. (Dimitar Demerdzhiev), I.A. and D.D. (Dobromir Dobrev); resources, D.D. (Dimitar Demerdzhiev) and I.A.; data curation, D.D. (Dimitar Demerdzhiev), I.A. and D.D. (Dobromir Dobrev); writing—original draft preparation, D.D. (Dimitar Demerdzhiev); writing—review and editing, I.A. and D.D. (Dobromir Dobrev); visualization, D.D. (Dimitar Demerdzhiev) and D.D. (Dobromir Dobrev); supervision, D.D. (Dimitar Demerdzhiev) and D.D. (Dobromir Dobrev); project administration, D.D. (Dimitar Demerdzhiev); funding acquisition, D.D. (Dimitar Demerdzhiev) and I.A. All authors have read and agreed to the published version of the manuscript.

**Funding:** This work was partially funded by the projects: "Conservation of globally important biodiversity in high nature value semi-natural grasslands through support for traditional local economy" (GEFSEC Project No 43595) and "Conservation of Imperial Eagle and Saker Falcon in key Natura 2000 sites in Bulgaria" (LIFE07 NAT/BG/000068).

**Institutional Review Board Statement:** Not applicable.

**Data Availability Statement:** Not applicable.

**Acknowledgments:** We would like to express our most sincere gratitude to Vladimir Trifonov, Atanas Demerdzhiev, Stoycho Stoychev, Girgina Daskalova, Nikolay Terziev, Georgi Pogeorgiev, Dimitar Plachiyski, Vladimir Dobrev, Kiril Metodiev, Nedko Nedialkov, Krasimira Demerdzhieva, Volen Arkumarev, Valentin Gochev, Georgi Georgiev, Vera Dyulgerska, Hristo Hristov, Hristo Ivanov, Svetoslav Spasov, Stefan Avramov, Marin Kurtev, and Milan Bakalov, who took part in the field work. Without their assistance, this survey would not be possible. We are grateful to tree anonymous reviewers who improved the draft of the manuscript.

**Conflicts of Interest:** The authors declare no conflict of interest.

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
