# Peer review of "Foraging Patterns of Non-Territorial Eastern Imperial Eagle (Aquila heliaca): A Case of Successful Adaptation"

_diversity, doi:10.3390/d14121060_

Round 1

Reviewer 1 Report

This manuscript represents a complete description of the feeding habits of floating EIE. In addition, the authors compare several aspects of the floater vs. adult feeding, also exploring the diet of floaters in detail. The manuscript is worth publishing, and I would add two suggestions: (1) the inclusion of further comparisons with tropical species in both the introduction and discussion, and (2) the remodeling of the theoretical background in the introduction, since optimal foraging died several years ago. These being solved, I would wholeheartedly recommend the manuscript for publication.

Author Response

Dear Reviewer,

thank you kindly for your comments and suggestions on our manuscript. We find them appropriate and hence we incorporated them in the manuscript.

Reviewer 1:

P1L42-43: the optimal foraging theory is defunct for decades. You should update it to something more modern, were risk and mating are incorporated into animal decision making. Please see:

Stephens, D. W., Brown, J. S., & Ydenberg, R. C. (Eds.). (2008). Foraging: behavior and ecology. University of Chicago Press.

Authors reply:

OK, we adopted your suggestion. See lines: 43-53.

Reviewer 1

P2L52-53: if you want an example from a tropical system, please see:

Miranda, E. B., Peres, C. A., Carvalho-Rocha, V., Miguel, B. V., Lormand, N., Huizinga, N., ... & Downs, C. T. (2021). Tropical deforestation induces thresholds of reproductive viability and habitat suitability in Earth’s largest eagles. Scientific reports, 11(1), 1-17.

P2L54-56: if you want an example from a tropical system, please see:

Miranda, E. B., Campbell-Thompson, E., Muela, A., & Vargas, F. H. (2018). Sex and breeding status affect prey composition of Harpy Eagles Harpia harpyja. Journal of Ornithology, 159(1), 141-150.

Miranda, E.B., Kenup, C.F., Campbell-Thompson, E., Vargas, F.H., Muela, A., Watson, R., Peres, C.A. and Downs, C.T., 2020. High moon brightness and low ambient temperatures affect sloth predation by harpy eagles. PeerJ, 8, p.e9756.

P2L58-63: please also explain how that differs from tropical species. You may like to learn from the following papers:

Zuluaga, S., Vargas, F. H., Aráoz, R., & Grande, J. M. (2022). Main aerial top predator of the Andean Montane Forest copes with fragmentation, but may be paying a high cost. Global Ecology and Conservation, e02174.

Muller, R., Amar, A., Sumasgutner, P., McPherson, S. C., & Downs, C. T. (2020). Urbanization is associated with increased breeding rate, but decreased breeding success, in an urban population of near-threatened African Crowned Eagles. The Condor, 122(3), duaa024.

McPherson, S. C., Brown, M., & Downs, C. T. (2019). Home range of a large forest eagle in a suburban landscape: crowned eagles (Stephanoaetus coronatus) in the Durban Metropolitan Open Space System, South Africa. Journal of Raptor Research, 53(2), 180-188.

Authors reply:

Thank you for your valuable suggestions. We checked these papers and referenced to the most appropriate one. Lines 74-75.

Reviewer 2 Report

I really enjoy reading the draft. Valuable article that provides novel information on the ecology of Aquila heliaca and its conservation. Very well written article. Very complete introduction, correct and well explained methodology, very extensive time series of field work, very complete discussion with well explained contens.

Author Response

Dear Reviewer,

thank you very much for your positive feedback and we are happy that you find it well structured and written.

Reviewer 3 Report

The authors have collected foraging data of Imperial Eagles in SE Bulgaria. with a stress on the floater population. This is a relatively unusual perspective and important to be published. The study is well thought out, the statistics appropriate, and the paper is reasonably well written. However,  it must be edited severely for language and context. Here are some examples for correction:

Title: Foraging patterns of non-territorial Eastern Imperial Eagle (Aquila heliaca): a case of successful adaptation

Abstract:

Line 13 - ……. birds of prey (floaters) are scarce and neglected. (same in line 55)

Give Latin names of all organisms - Eastern Imperial Eagle, European Souslik, Hedgehog

Introduction:

Lines 76-78. Also see – Hadad et al. 2022. Sustaining winter raptor populations in Central Israel: a 38 year perspective. Sustainability 14:12481. https://doi.org/10.3390/su141912481.

Line 84  - ….. we built a simple model of optimal foraging….

Line 90 - ……. to optimize foraging; - Is it? I think it is to maximize energy intake with minimal expenditure?

Materials & Methods

Line 100 - …… during 25 years’ period (1998- ……..

Line 103 - Most of them (n = 147; 79%) were ……….

Line 104 - ……….. , which accumulated dozens of floaters yearly … - reword

Line 109 - ……. After localization of identifying the roost…….

Figure 1. Give Latin name

Line 124 - Eagle plumage was given by Forsman (2005). – reword

Line 136 - ………. behavior was given described by Watson……

Author Response

Dear Reviewer,

thank you for your comments. They helped a lot and improved the text significantly. Upon to your suggestion, the manuscript was edited by a native speaker as well. You will find our answers attached below.

Reviewer 3:

The authors have collected foraging data of Imperial Eagles in SE Bulgaria. with a stress on the floater population. This is a relatively unusual perspective and important to be published. The study is well thought out, the statistics appropriate, and the paper is reasonably well written. However,  it must be edited severely for language and context. Here are some examples for correction:

Title: Foraging patterns of non-territorial Eastern Imperial Eagle (Aquila heliaca): a case of successful adaptation

Authors reply:

Thank you for your positive reply and the positive evaluation of our manuscript. The manuscript was edited by a native speaker and amended where necessary.

Our native speaker editor checked the title and finds it grammatically correct. In our view it is well corresponding the subject of the paper.

Reviewer 3:

Abstract:

Line 13 - ……. birds of prey (floaters) are scarce and neglected. (same in line 55)

Authors reply:

Ok, we removed it and rephrased the sentence accordingly.

Reviewer 3:

Give Latin names of all organisms - Eastern Imperial Eagle, European Souslik, Hedgehog

Authors reply:  According to the journal requirements this is not allowed for the abstract section but at the time of their first appearance in the text. We did so.

Reviewer 3:

Introduction:

Lines 76-78. Also see – Hadad et al. 2022. Sustaining winter raptor populations in Central Israel: a 38 year perspective. Sustainability 14:12481. https://doi.org/10.3390/su141912481.

Authors reply:  

We added the reference.

Reviewer 3:

Line 84  - ….. we built a simple model of optimal foraging….

Authors reply:

We amended the text according to your suggestion.

Reviewer 3:

Line 90 - ……. to optimize foraging; - Is it? I think it is to maximize energy intake with minimal expenditure?

Authors reply:  

Yes, we meant this and we think this is essentially what foraging optimization means.

Reviewer 3:

Materials & Methods

Line 100 - …… during 25 years’ period (1998- ……..

Authors reply:  

Rephrased accordingly.

Reviewer 3:

Line 103 - Most of them (n = 147; 79%) were ……….

Authors reply:  

The native speaker advised us and we modified it accordingly.

Reviewer 3:

Line 104 - ……….. , which accumulated dozens of floaters yearly … - reword

Authors reply:  

We reworded it accordingly.

Reviewer 3:

Line 109 - ……. After localization of identifying the roost…….

Authors reply:  

We reworded it accordingly.

Reviewer 3:

Figure 1. Give Latin name

Authors reply:  

Done.

Reviewer 3:

Line 124 - Eagle plumage was given by Forsman (2005). – reword

Authors reply:  

Done.

Reviewer 3:

Line 136 - ………. behavior was given described by Watson……

Authors reply:  

Done.

Round 2

Reviewer 3 Report

The authors have been minimalistic in their editing. However, the paper is important enough that the wrinkles/edits will be ironed out in the editorial process.